# AttentionXML: Label Tree-based Attention-Aware Deep Model for High-Performance Extreme Multi-Label Text Classification

**Ronghui You**[1], **Zihan Zhang**[1], **Ziye Wang**[2], **Suyang Dai**[1],
**Hiroshi Mamitsuka**[5,6], **Shanfeng Zhu**[1,3,4,*]

[1] Shanghai Key Lab of Intelligent Information Processing, School of Computer Science,
[2] Centre for Computational Systems Biology, School of Mathematical Sciences,
[3] Shanghai Institute of Artificial Intelligence Algorithms and ISTBI,
[4] Key Lab of Computational Neuroscience and Brain-Inspired Intelligence (MOE),
Fudan University, Shanghai, China;
[5] Bioinformatics Center, Institute for Chemical Research, Kyoto University, Japan;
[6] Department of Computer Science, Aalto University, Espoo and Helsinki, Finland
`{rhyou18,zhangzh17,zywang17,sydai16}@fudan.edu.cn`
`mami@kuicr.kyoto-u.ac.jp, zhusf@fudan.edu.cn`

## Abstract

Extreme multi-label text classification (XMTC) is an important problem in the era of *big data*, for tagging a given text with the most relevant multiple labels from an extremely large-scale label set. XMTC can be found in many applications, such as item categorization, web page tagging, and news annotation. Traditionally most methods used bag-of-words (BOW) as inputs, ignoring word context as well as deep semantic information. Recent attempts to overcome the problems of BOW by deep learning still suffer from 1) failing to capture the important subtext for each label and 2) lack of scalability against the huge number of labels. We propose a new label tree-based deep learning model for XMTC, called AttentionXML, with two unique features: 1) a multi-label attention mechanism with raw text as input, which allows to capture the most relevant part of text to each label; and 2) a shallow and wide probabilistic label tree (PLT), which allows to handle millions of labels, especially for "tail labels". We empirically compared the performance of AttentionXML with those of eight state-of-the-art methods over six benchmark datasets, including Amazon-3M with around 3 million labels. AttentionXML outperformed all competing methods under all experimental settings. Experimental results also show that AttentionXML achieved the best performance against tail labels among label tree-based methods. The code and datasets are available at `http://github.com/yourh/AttentionXML` .

## 1 Introduction

Extreme multi-label text classification (XMTC) is a natural language processing (NLP) task for tagging each given text with its most relevant multiple labels from an extremely large-scale label set. XMTC predicts multiple labels for a text, which is different from multi-class classification, where each instance has only one associated label. Recently, XMTC has become increasingly important, due to the fast growth of the data scale. In fact, over hundreds of thousands, even millions of labels and samples can be found in various domains, such as item categorization in e-commerce, web page tagging, news annotation, to name a few. XMTC poses great computational challenges for developing effective and efficient classifiers with limited computing resource, such as an extremely large number of samples/labels and a large number of "tail labels" with very few positive samples.

Many methods have been proposed for addressing the challenges of XMTC. They can be categorized into the following four types: 1) 1-vs-All [3,4,30,31], 2) Embedding-based [7,27], 3) Instance [10,25] or label tree-based [11, 13, 24, 28]) and 4) Deep learning-based methods [17] (see Appendix for more descriptions on these methods). The most related methods to our work are deep learning-based and label tree-based methods. A pioneering deep learning-based method is XML-CNN [17], which uses a convolutional neural network (CNN) and dynamic pooling to learn the text representation. XML-CNN however cannot capture the most relevant parts of the input text to each label, because the same text representation is given for all labels. Another type of deep learning-based methods is sequence-to-sequence (Seq2Seq) learning-based methods, such as MLC2Seq [21], SGM [29] and SU4MLC [15]. These Seq2Seq learning-based methods use a recurrent neural network (RNN) to encode a given raw text and an attentive RNN as a decoder to generate predicted labels sequentially. However the underlying assumption of these models is not reasonable since in reality there are no orders among labels in multi-label classification. In addition, the requirement of extensive computing in the existing deep learning-based methods makes it unbearable to deal with datasets with millions of labels.

To handle such extreme-scale datasets, label tree-based methods use a probabilistic label tree (PLT) [11] to partition labels, where each leaf in PLT corresponds to an original label and each internal node corresponds to a pseudo-label (meta-label). Then by maximizing a lower bound approximation of the log likelihood, each linear binary classifier for a tree node can be trained independently with only a small number of relevant samples [24]. Parabel [24] is a state-of-the-art label tree-based method using bag-of-words (BOW) features. This method constructs a binary balanced label tree by recursively partitioning nodes into two balanced clusters until the cluster size (the number of labels in each cluster) is less than a given value (e.g. 100). This produces a "deep" tree (with a high tree depth) for an extreme scale dataset, which deteriorates the performance due to an inaccurate approximation of likelihood, and the accumulated and propagated errors along the tree. In addition, using balanced clustering with a large cluster size, many tail labels are combined with other dissimilar labels and grouped into one cluster. This reduces the classification performance on tail labels. On the other hand, another PLT-based method EXTREMETEXT [28], which is based on FASTTEXT [12], uses dense features instead of BOW. Note that EXTREMETEXT ignores the order of words without considering context information, which underperforms Parabel.

We propose a label tree-based deep learning model, AttentionXML, to address the current challenges of XMTC. AttentionXML uses raw text as its features with richer semantic context information than BOW features. AttentionXML is expected to achieve a high accuracy by using a BiLSTM (bidirectional long short-term memory) to capture long-distance dependency among words and a multi-label attention to capture the most relevant parts of texts to each label. Most state-of-the-art methods, such as DiSMEC [3] and Parabel [24], used only one representation for all labels including many dissimilar (unrelated) tail labels. It is difficult to satisfy so many dissimilar labels by the same representation. With multi-label attention, AttentionXML represents a given text differently for each label, which is especially helpful for many tail labels. In addition, by using a shallow and wide PLT and a top-down level-wise model training, AttentionXML can handle extreme-scale datasets. Most recently, Bonsai [13] also uses shallow and diverse PLTs by removing the balance constraint in the tree construction, which improves the performance by Parabel. Bonsai, however, needs high space complexity, such as a 1TB memory for extreme-scale datasets, because of using linear classifiers. Note that we conceive our idea that is independent from Bonsai, and apply it in deep learning based method using deep semantic features other than BOW features used in Bonsai. The experimental results over six benchmarks datasets including Amazon-3M [19] with around 3 million labels and 2 millions samples show that AttentionXML outperformed other state-of-the-art methods with competitive costs on time and space. The experimental results also show that AttentionXML is the best label tree-based method against tail labels.

## 2  AttentionXML

### 2.1  Overview

The main steps of AttentionXML are: (1) building a shallow and wide PLT (Figs. 1a and 1b); and (2) for each level $d$ $(d > 0)$ of a given constructed PLT, training an attention-aware deep model AttentionXML$_d$ with a BiLSTM and a multi-label attention (Fig. 1c). The pseudocodes for constructing PLT, training and prediction of AttentionXML are presented in **Appendix**.

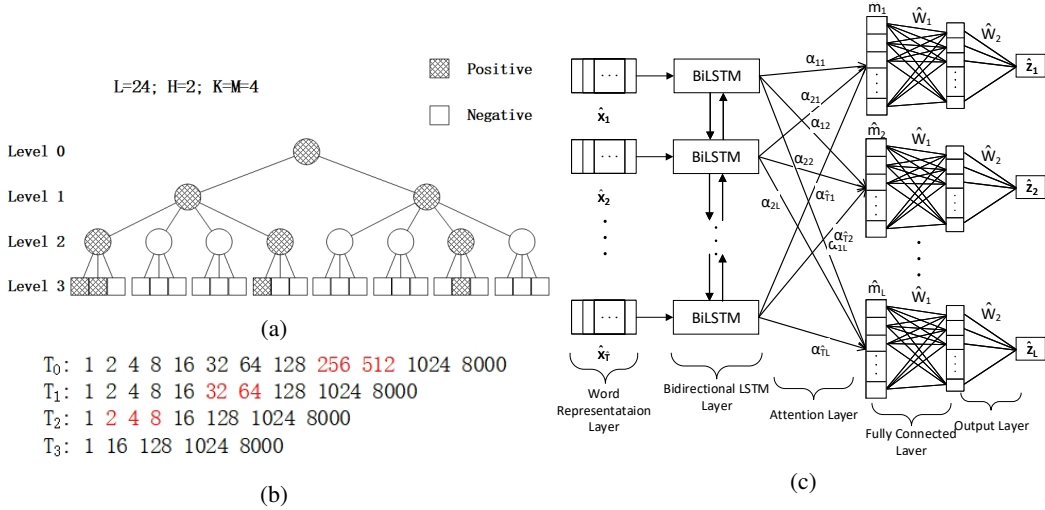

Figure 1: Label tree-based deep model AttentionXML for XMTC. (a) An example of PLT used in AttentionXML. (b) An example of a PLT building process with settings of $K = M = 8 = 2^3$ and $H = 3$ for $L = 8000$. The numbers from left to right show those of nodes for each level from top to down. The numbers in red show those of nodes in $T_h$ that are removed in order to obtain $T_{h+1}$. (c) Overview of attention-aware deep model in AttentionXML with text (length $\hat{T}$) as its input and predicted scores $\hat{z}$ as its output. The $\hat{x}_i \in \mathbb{R}^{\hat{D}}$ is the embeddings of $i$-th word(where $\hat{D}$ is the dimension of embeddings), $\alpha \in \mathbb{R}^{\hat{T} \times L}$ are the attention coefficients and $\hat{W}_1$ and $\hat{W}_2$ are parameters of the fully connected layer and output layer.

## 2.2 Building Shallow and Wide PLT

PLT [10] is a tree with $L$ leaves where each leaf corresponds to an original label. Given a sample $x$, we assign a label $z_n \in \{0, 1\}$ for each node $n$, which indicates whether the subtree rooted at node $n$ has a leaf (original label) relevant to this sample. PLT estimates the conditional probability $P(z_n | z_{Pa(n)} = 1, x)$ to each node $n$. The marginal probability $P(z_n = 1 | x)$ for each node $n$ can be easily derived as follows by the chain rule of probability:

$$P(z_n = 1 | x) = \prod_{i \in Path(n)} P(z_i = 1 | z_{Pa(i)} = 1, x) \tag{1}$$

where $Pa(n)$ is the parent of node $n$ and $Path(n)$ is the set of nodes on the path from node $n$ to the root (excluding the root).

As mentioned in **Introduction**, large tree height $H$ (excluding the root and leaves) and large cluster size $M$ will harm the performance. So in AttentionXML, we build a shallow (a small $H$) and wide (a small $M$) PLT $T_H$. First, we built an initial PLT, $T_0$, by a top-down hierarchical clustering, which was used in Parabel [24], with a small cluster size $M$. In more detail, we represent each label by normalizing the sum of BOW features of text annotated by this label. The labels are then recursively partitioned into two smaller clusters, which correspond to internal tree nodes, by a balanced $k$-means ($k$=2) until the number of labels smaller than $M$ [24]. $T_0$ is then compressed into a shallow and wide PLT, i.e. $T_H$, which is a $K(= 2^c)$ ways tree with the height of $H$. This compress operation is similar to the pruning strategy in some hierarchical multi-class classification methods [1, 2]. We first choose all parents of leaves as $S_0$ and then conduct compress operations $H$ times, resulting in $T_H$. The compress operation has three steps: for example in the $h$-th compress operation over $T_{h-1}$, we (1) choose $c$-th ancestor nodes ($h < H$) or the root ($h = H$) as $S_h$, (2) remove nodes between $S_{h-1}$ and $S_h$, and (3) then reset nodes in $S_h$ as parents of corresponding nodes in $S_{h-1}$. This finally results in a shallow and wide tree $T_H$. Practically we use $M = K$ so that each internal node except the root has no more than $K$ children. Fig 1b shows an example of building PLT. More examples can be found in **Appendix**.

### 2.3 Learning AttentionXML

Given a built PLT, training a deep model against nodes at a deeper level is more difficult because nodes at a deeper level have less positive examples. Training a deep model for all nodes of different levels together is hard to optimize and harms the performance, which can only speed up marginally. Thus we train AttentionXML in a level-wise manner as follows:

1. AttentionXML trains a single deep model for each level of a given PLT in a top-down manner. Note that labeling each level of the PLT is still a multi-label classification problem. For the nodes of first level (children of the root), AttentionXML (named AttentionXML$_1$ for the first level) can be trained for these nodes directly.

2. AttentionXML$_d$ for the $d$-th level ($d > 1$) of the given PLT is only trained by candidates $g(x)$ for each sample $x$. Specifically, we sort nodes of the $(d-1)$-th level by $z_n$ (from positives to negatives) and then their scores predicted by AttentionXML$_{d-1}$ in the descending order. We keep the top $C$ nodes at the $(d-1)$-th level and choose their children as $g(x)$. It's like a kind of additional negative sampling and we can get a more precise approximation of log likelihood than only using nodes with positive parents.

3. During prediction, for the $i$-th sample, the predicted score $\hat{y}_{ij}$ for $j$-th label can be computed easily based on the probability chain rule. For the prediction efficiency, we use beam search [13, 24]: for the $d$-th ($d > 1$) level we only predict scores of nodes that belong to nodes of the $(d-1)$-th level with top $C$ predicted scores.

We can see that the deep model without using a PLT can be regarded as a special case of AttentionXML with a PLT with only the root and $L$ leaves.

### 2.4 Attention-Aware Deep Model

Attention-aware deep model in AttentionXML consists of five layers: 1) Word Representation Layer, 2) Bidirectional LSTM Layer, 3) Multi-label Attention Layer, 4) Fully Connected Layer and 5) Output Layer. Fig. 1c shows a schematic picture of attention-aware deep model in AttentionXML.

#### 2.4.1 Word Representation Layer

The input of AttentionXML is raw tokenized text with length $\hat{T}$. Each word is represented by a deep semantic dense vector, called *word embedding* [22]. In our experiments, we use pre-trained 300-dimensional GloVe [22] word embedding as our initial word representation.

#### 2.4.2 Bidirectional LSTM Layer

RNN is a type of neural network with a memory state to process sequence inputs. Traditional RNN has a problem called *gradient vanishing and exploding* during training [6]. Long short-term memory (LSTM) [8] is proposed for solving this problem. We use a Bidirectional LSTM (BiLSTM) to capture both the left- and right-sides context (Fig. 1c), where at each time step $t$ the output $\hat{\mathbf{h}}_t$ is obtained by concatenating the forward output $\overrightarrow{\mathbf{h}}_t$ and the backward output $\overleftarrow{\mathbf{h}}_t$.

#### 2.4.3 Multi-Label Attention

Recently, an *attention mechanism* in neural networks has been successfully used in many NLP tasks, such as machine translation, machine comprehension, relation extraction, and speech recognition [5, 18]. The most relevant context to each label can be different in XMTC. AttentionXML computes the (linear) combination of context vectors $\hat{\mathbf{h}}_i$ for each label through a *multi-label attention mechanism*, inspired by [16], to capture various intensive parts of a text. That is, the output of multi-label attention layer $\hat{\mathbf{m}}_j \in \mathbb{R}^{2\hat{N}}$ of the $j$-th label can be obtained as follows:

$$\hat{\mathbf{m}}_j = \sum_{i=1}^{\hat{T}} \alpha_{ij}\hat{\mathbf{h}}_i, \qquad \alpha_{ij} = \frac{e^{\hat{\mathbf{h}}_i\hat{\mathbf{w}}_j}}{\sum_{t=1}^{\hat{T}} e^{\hat{\mathbf{h}}_t\hat{\mathbf{w}}_j}}, \tag{2}$$

where $\alpha_{ij}$ is the normalized coefficient of $\hat{\mathbf{h}}_i$ and $\hat{\mathbf{w}}_j \in \mathbb{R}^{2\hat{N}}$ is the so-called attention parameters. Note that $\hat{w}_j$ is different for each label.

Table 1: Datasets we used in our experiments.

| Dataset | $N_{train}$ | $N_{test}$ | $D$ | $L$ | $\overline{L}$ | $\hat{L}$ | $\overline{W}_{train}$ | $\overline{W}_{test}$ |
|---|---|---|---|---|---|---|---|---|
| EUR-Lex | 15,449 | 3,865 | 186,104 | 3,956 | 5.30 | 20.79 | 1248.58 | 1230.40 |
| Wiki10-31K | 14,146 | 6,616 | 101,938 | 30,938 | 18.64 | 8.52 | 2484.30 | 2425.45 |
| AmazonCat-13K | 1,186,239 | 306,782 | 203,882 | 13,330 | 5.04 | 448.57 | 246.61 | 245.98 |
| Amazon-670K | 490,449 | 153,025 | 135,909 | 670,091 | 5.45 | 3.99 | 247.33 | 241.22 |
| Wiki-500K | 1,779,881 | 769,421 | 2,381,304 | 501,008 | 4.75 | 16.86 | 808.66 | 808.56 |
| Amazon-3M | 1,717,899 | 742,507 | 337,067 | 2,812,281 | 36.04 | 22.02 | 104.08 | 104.18 |

$N_{train}$: #training instances, $N_{test}$: #test instances, $D$: #features, $L$: #labels, $\overline{L}$: average #labels per instance, $\hat{L}$: the average #instances per label, $\overline{W}_{train}$: the average #words per training instance and $\overline{W}_{test}$: the average #words per test instance. The partition of training and test is from the data source.

### 2.4.4 Fully Connected and Output Layer

AttentionXML has one (or two) fully connected layers and one output layer. The same parameter values are used for all labels at the fully connected (and output) layers, to emphasize differences of attention among all labels. Also sharing the parameter values of fully connected layers among all labels can largely reduce the number of parameters to avoid overfitting and keep the model scale small.

### 2.4.5 Loss Function

AttentionXML uses the binary cross-entropy loss function, which is used in XML-CNN [17] as the loss function. Since the number of labels for each instance varies, we do not normalize the predicted probability which is done in multi-class classification.

### 2.5 Initialization on parameters of AttentionXML

We initialize the parameters of AttentionXML$_d$ ($d > 1$) by using the parameters of trained AttentionXML$_{d-1}$, except the attention layers. This initialization helps models of deeper levels converge quickly, resulting in improvement of the final accuracy.

### 2.6 Complexity Analysis

The deep model without a PLT is hard to deal with extreme-scale datasets, because of high time and space complexities of the multi-label attention mechanism. Multi-label attention in the deep model needs $O(BL\hat{N}\hat{T})$ time and $O(BL(\hat{N}+\hat{T}))$ space for each batch iteration, where $B$ is the batch size. For large number of labels ($L > 100$k), the time cost is huge. Also the whole model cannot be saved in the limited memory space of GPUs. On the other hand, the time complexity of AttentionXML with a PLT is much smaller than that without a PLT, although we need train $H + 1$ different deep models. That is, the label size of AttentionXML$_1$ is only $L/K^H$, which is much smaller than $L$. Also the number of candidate labels of AttentionXML$_d(d > 1)$ is only $C \times K$, which is again much smaller than $L$. Thus our efficient label tree-based AttentionXML can be run even with the limited GPU memory.

## 3 Experimental Results

### 3.1 Dataset

We used six most common XMTC benchmark datasets (Table 1): three large-scale datasets ($L$ ranges from 4K to 30K) : EUR-Lex[1] [20], Wiki10-31K[2] [32], and AmazonCat-13K [2] [19]; and three extreme-scale datasets ($L$ ranges from 500K to 3M): Amazon-670K[2] [19], Wiki-500K[2] and Amazon-3M[2] [19]. Note that both Wiki-500K and Amazon-3M have around two million samples for training.

Table 2: Hyperparameters we used in our experiments, practical computation time and model size.

| Datasets | $E$ | $B$ | $\hat{N}$ | $\hat{N}_{fc}$ | $H$ | $M = K$ | $C$ | Train (hours) | Test (ms/ sample) | Model Size (GB) |
|---|---|---|---|---|---|---|---|---|---|---|
| EUR-Lex | 30 | 40 | 256 | 256 | - | - | - | 0.51 | 2.07 | 0.20 |
| Wiki10-31K | 30 | 40 | 256 | 256 | - | - | - | 1.27 | 4.53 | 0.62 |
| AmazonCat-13K | 10 | 200 | 512 | 512,256 | - | - | - | 13.11 | 1.63 | 0.63 |
| Amazon-670K | 10 | 200 | 512 | 512,256 | 3 | 8 | 160 | 13.90 | 5.27 | 5.52 |
| Wiki-500K | 5 | 200 | 512 | 512,256 | 1 | 64 | 15 | 19.55 | 2.46 | 3.11 |
| Amazon-3M | 5 | 200 | 512 | 512,256 | 3 | 8 | 160 | 31.67 | 5.92 | 16.14 |

$E$: The number of epoch; $B$: The batch size; $N$: The hidden unit size of LSTM; $N_{fc}$: The hidden unit size of fully connected layers; $H$: The height of PLT (excluding the root and leaves); $M$: The maximum cluster size; $K$: The parameters of the compress process, and here we set $M = K = 2^c$; $C$: The number of parents of candidate nodes.

## 3.2 Evaluation Measures

We chose P@$k$ (Precision at $k$) [10] as our evaluation metrics for performance comparison, since $P@k$ is widely used for evaluating the methods for XMTC.

$$P@k = \frac{1}{k} \sum_{l=1}^{k} \mathbf{y}_{rank(l)} \tag{3}$$

where $\mathbf{y} \in \{0, 1\}^L$ is the true binary vector, and $rank(l)$ is the index of the $l$-th highest predicted label. Another common evaluation metric is $N@k$ (normalized Discounted Cumulative Gain at $k$). Note that $P@1$ is equivalent to $N@1$. We evaluated performance by $N@k$, and confirmed that the performance of $N@k$ kept the same trend as $P@k$. We thus omit the results of $N@k$ in the main text due to space limitation (see **Appendix**).

## 3.3 Competing Methods and Experimental Settings

We compared the state-of-the-art and most representative XMTC methods (implemented by the original authors) with AttentionXML: AnnexML[3] (embedding), DiSMEC[4] (1-vs-All), MLC2Seq[5] (deep learning), XML-CNN[2] (deep learning), PfastreXML[2] (instance tree), Parabel[2] (label tree) and XT[6] (ExtremeText) (label tree) and Bonsai[7] (label tree).

For each dataset, we used the most frequent words in the training set as a limited-size vocabulary (not over 500,000). Word embeddings were fine-tuned during training except EUR-Lex and Wiki10-31K. We truncated each text after 500 words for training and predicting efficiently. We used dropout [26] to avoid overfitting after the embedding layer with the drop rate of 0.2 and after the BiLSTM with the drop rate of 0.5. Our model was trained by Adam [14] with the learning rate of 1e-3. We also used SWA (stochastic weight averaging) [9] with a constant learning rate to enhance the performance. We used a three PLTs ensemble in AttentionXML similar to Parabel [23] and Bonsai [13]. We also examined performance of AttentionXML with only one PLT (without ensemble), called AttentionXML-1. On three large-scale datasets, we used AttentionXML with a PLT including only a root and $L$ leaves(which can also be considered as the deep model without PLTs). Other hyperparameters in our experiments are shown in Tabel 2.

## 3.4 Performance comparison

Table 3 shows the performance results of AttentionXML and other competing methods by $P@k$ over all six benchmark datasets. Following the previous work on XMTC, we focus on top predictions by varying $k$ at 1, 3 and 5 in $P@k$, resulting in 18 (= three k × six datasets) values of $P@k$ for each method.

Table 3: Performance comparisons of AttentionXML and other competing methods over six benchmarks. The results with the stars are from **Extreme Classification Repository** directly.

| Methods | P@1=N@1 | P@3 | P@5 | Methods | P@1=N@1 | P@3 | P@5 |
|---|---|---|---|---|---|---|---|
| | EUR-Lex | | | | Amazon-670K | | |
| AnnexML | 79.66 | 64.94 | 53.52 | AnnexML | 42.09 | 36.61 | 32.75 |
| DiSMEC | 83.21 | 70.39 | 58.73 | DiSMEC | 44.78 | 39.72 | 36.17 |
| PfastreXML | 73.14 | 60.16 | 50.54 | PfastreXML* | 36.84 | 34.23 | 32.09 |
| Parabel | 82.12 | 68.91 | 57.89 | Parabel | 44.91 | 39.77 | 35.98 |
| XT | 79.17 | 66.80 | 56.09 | XT | 42.54 | 37.93 | 34.63 |
| Bonsai | 82.30 | 69.55 | 58.35 | Bonsai | 45.58 | 40.39 | 36.60 |
| MLC2Seq | 62.77 | 59.06 | 51.32 | MCL2Seq | - | - | - |
| XML-CNN | 75.32 | 60.14 | 49.21 | XML-CNN | 33.41 | 30.00 | 27.42 |
| AttentionXML-1 | *85.49* | *73.08* | *61.10* | AttentionXML-1 | *45.66* | *40.67* | *36.94* |
| AttentionXML | **87.12** | **73.99** | **61.92** | AttentionXML | **47.58** | **42.61** | **38.92** |
| | Wiki10-31K | | | | Wiki-500K | | |
| AnnexML | 86.46 | 74.28 | 64.20 | AnnexML | 64.22 | 43.15 | 32.79 |
| DiSMEC | 84.13 | 74.72 | 65.94 | DiSMEC | 70.21 | 50.57 | 39.68 |
| PfastreXML* | 83.57 | 68.61 | 59.10 | PfastreXML | 56.25 | 37.32 | 28.16 |
| Parabel | 84.19 | 72.46 | 63.37 | Parabel | 68.70 | 49.57 | 38.64 |
| XT | 83.66 | 73.28 | 64.51 | XT | 65.17 | 46.32 | 36.15 |
| Bonsai | 84.52 | 73.76 | 64.69 | Bonsai | 69.26 | 49.80 | 38.83 |
| MLC2Seq | 80.79 | 58.59 | 54.66 | MCL2Seq | - | - | - |
| XML-CNN | 81.41 | 66.23 | 56.11 | XML-CNN | - | - | - |
| AttentionXML-1 | *87.05* | *77.78* | *68.78* | AttentionXML-1 | *75.07* | *56.49* | *44.41* |
| AttentionXML | **87.47** | **78.48** | **69.37** | AttentionXML | **76.95** | **58.42** | **46.14** |
| | AmazonCat-13K | | | | Amazon-3M | | |
| AnnexML | 93.54 | 78.36 | 63.30 | AnnexML | *49.30* | 45.55 | 43.11 |
| DiSMEC | 93.81 | 79.08 | 64.06 | DiSMEC* | 47.34 | 44.96 | 42.80 |
| PfastreXML* | 91.75 | 77.97 | 63.68 | PfastreXML* | 43.83 | 41.81 | 40.09 |
| Parabel | 93.02 | 79.14 | 64.51 | Parabel | 47.42 | 44.66 | 42.55 |
| XT | 92.50 | 78.12 | 63.51 | XT | 42.20 | 39.28 | 37.24 |
| Bonsai | 92.98 | 79.13 | 64.46 | Bonsai | 48.45 | 45.65 | 43.49 |
| MLC2Seq | 94.29 | 69.45 | 57.55 | MCL2Seq | - | - | - |
| XML-CNN | 93.26 | 77.06 | 61.40 | XML-CNN | - | - | - |
| AttentionXML-1 | *95.65* | *81.93* | *66.90* | AttentionXML-1 | 49.08 | *46.04* | *43.88* |
| AttentionXML | **95.92** | **82.41** | **67.31** | AttentionXML | **50.86** | **48.04** | **45.83** |

1) AttentionXML (with a three PLTs ensemble) outperformed all eight competing methods by $P@k$. For example, for $P@5$, among all datasets, AttentionXML is at least $4\%$ higher than the second best method (Parabel on AmazonCat-13K). For Wiki-500K, AttentionXML is even more than $17\%$ higher than the second best method (DiSMEC). AttentionXML also outperformed AttentionXML-1 (without ensemble), especially on three extreme-scale datasets. That's because on extreme-scale datasets, the ensemble with different PLTs reduces more variance, while on large-scale datasets models the ensemble is with the same PLTs (only including the root and leaves). Note that AttentionXML-1 is much more efficient than AttentionXML, because it only trains one model without ensemble.

2) AttentionXML-1 outperformed all eight competing methods by $P@k$, except only one case. Performance improvement was especially notable for EUR-Lex, Wiki10-31K and Wiki-500K, with longer texts than other datasets (see Table 1). For example, for $P@5$, AttentionXML-1 achieved 44.41, 68.78 and 61.10 on Wiki-500K, Wiki10-31K and EUR-Lex, which were around $12\%$, $4\%$ and $4\%$ higher than the second best, DiSMEC with 39.68, 65.94 and 58.73, respectively. This result highlights that longer text has larger amount of context information, where multi-label attention can focus more on the most relevant parts of text and extract the most important information on each label.

3) Parabel, a method using PLTs, can be considered as taking the advantage of both tree-based (PfastreXML) and 1-vs-All (DiSMEC) methods. It outperformed PfastreXML and achieved a similar performance to DiSMEC (which is however much more inefficient). ExtremeText (XT) is an online

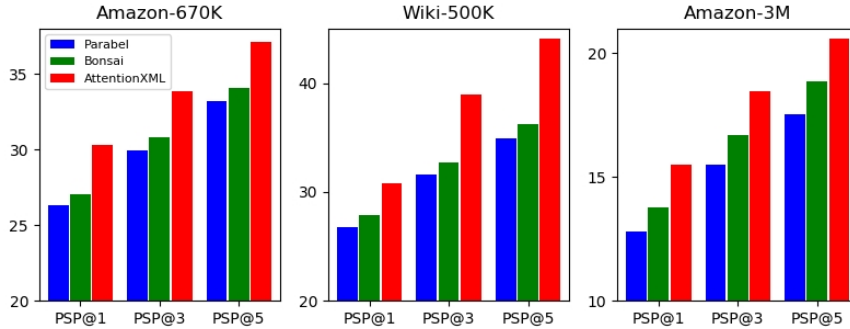

Figure 2: $PSP@k$ of label tree-based methods.

learning method with PLTs (similar to Parabel), which used dense instead of sparse representations and achieved slightly lower performance than parabel. Bonsai, another method using PLTs, outperformed Parabel on all datasets except AmazonCat-13K. In addition, Bonsai achieved better performance than DiSMEC on Amazon-670K and Amazon-3M. This result indicates that the shallow and diverse PLTs in Bonsai improves its performance. However, Bonsai needs much more memory than Parabel, for example, 1TB memory for extreme-scale datasets. Note that AttentionXML-1 with only one shallow and wide PLT, still significantly outperformed both Parabel and Bonsai on all extreme-scale datasets, especially Wiki-500K.

4) MLC2Seq, a deep learning-based method, obtained the worst performance on the three large-scale datasets, probably because of its unreasonable assumption. XML-CNN, another deep learning-based method with a simple dynamic pooling was much worse than the other competing methods, except MLC2Seq. Note that both MLC2Seq and XML-CNN are unable to deal with datasets with millions of labels.

5) AttentionXML was the best method among all the competing methods, on the three extreme-scale datasets (Amazon-670K, Wiki-500K and Amazon-3M). Although the improvement by AttentionXML-1 over the second and third best methods (Bonsai and DiSMEC) is rather slight, AttentionXML-1 is much faster than DiSMEC and uses much less memory than Bonsai. In addition, the improvement by AttentionXML with a three PLTs ensemble over Bonsai and DiSMEC is more significant, which is still faster than DiSMEC and uses much less memory than Bonsai.

6) AnnexML, the state-of-the-art embedding-based method, reached the second best $P@1$ on Amazon-3M and Wiki10-31K, respectively. Note that the performance of AnnexML was not necessarily so on the other datasets. The average number of labels per sample of Amazon-3M (36.04) and Wiki10-31K (18.64) is several times larger than those of other datasets (only around 5). This suggests that each sample in these datasets has been well annotated. Under this case, embedding-based methods may acquire more complete information from the nearest samples by using KNN (k-nearset neighbors) and might gain a relatively good performance on such datasets.

### 3.5 Performance on tail labels

We examined the performance on tail labels by $PSP@k$ (propensity scored precision at k) [10]:

$$\text{PSP@}k = \frac{1}{k} \sum_{l=1}^{k} \frac{\mathbf{y}_{\text{rank}(l)}}{\mathbf{p}_{\text{rank}(l)}} \tag{4}$$

where $\mathbf{p}_{\text{rank}(l)}$ is the propensity score [10] of label $rank(l)$. Fig 2 shows the results of three label tree-based methods (Parabel, Bonsai and AttentionXML) on the three extreme-scale datasets. Due to space limitation, we reported $PSP@k$ results of AttentionXML and all compared methods including ProXML [4] (a state-of-the-art method on $PSP@k$) on six benchmarks in **Appendix**.

AttentionXML outperformed both Parabel and Bonsai in $PSP@k$ on all datasets. AttentionXML use a shallow and wide PLT, which is different from Parabel. Thus this result indicates that this shallow and wide PLT in AttentionXML is promising to improve the performance on tail labels. Additionally, multi-label attention in AttentionXML would be also effective for tail labels, because of capturing

Table 4: P@5 of XML-CNN, BiLSTM and AttentionXML (all without ensemble)

| Dataset | XML-CNN | BiLSTM | AttentionXML (BiLSTM+Att) | AttentionXML (BiLSTM+Att+SWA) |
|---|---|---|---|---|
| EUR-Lex | 49.21 | 53.12 | 59.61 | **61.10** |
| Wiki10-31K | 56.21 | 59.55 | 66.51 | **68.78** |
| AmazonCat-13K | 61.40 | 63.57 | 66.29 | **66.90** |

Table 5: Performance of variant number of trees in AttentionXML.

| Trees | Amazon-670K | | | Wiki-500K | | | Amazon-3M | | |
|---|---|---|---|---|---|---|---|---|---|
| | P@1 | P@3 | P@5 | P@1 | P@3 | P@5 | P@1 | P@3 | P@5 |
| 1 | 45.66 | 40.67 | 36.94 | 75.07 | 56.49 | 44.41 | 49.08 | 46.04 | 43.88 |
| 2 | 46.86 | 41.95 | 38.27 | 76.44 | 57.92 | 45.68 | 50.34 | 47.45 | 45.26 |
| 3 | 47.58 | 42.61 | 38.92 | 76.95 | 58.42 | 46.14 | 50.86 | 48.04 | 45.83 |
| 4 | **48.03** | **43.05** | **39.32** | **77.21** | **58.72** | **46.40** | **51.66** | **48.39** | **46.23** |

the most important parts of text for each label, while Bonsai uses just the same BOW features for all labels.

### 3.6 Ablation Analysis

For examining the impact of BiLSTM and multi-label attention, we also run a model which consists of a BiLSTM, a max-pooling (instead of the attention layer of AttentionXML), and the fully connected layers (from XML-CNN). Tabel 4 shows the $P@5$ results on three large-scale datasets. BiLSTM outperformed XML-CNN on all three datasets, probably because of capturing the long-distance dependency among words. AttentionXML (BiLSTM+Attn) further outperformed XML-CNN and BiLSTM, especially on EUR-Lex and Wiki10-31K, which have long texts. Comparing with a simple dynamic pooling, obviously multi-label attention can extract the most important information to each label from long texts more easily. In addition, Table 4 shows that SWA has a favorable effect on improving prediction accuracy.

### 3.7 Impact of Number of PLTs in AttentionXML

We examined the performance of ensemble with different number of PLTs in AttentionXML. Table 5 shows the performance comparison of AttentionXML with different number of label trees. We can see that more trees much improve the prediction accuracy. However, using more trees needs much more time for both training and prediction. So its a trade-off between performance and time cost.

### 3.8 Computation Time and Model Size

AttentionXML runs on 8 Nvidia GTX 1080Ti GPUs. Table 2 shows the computation time for training (hours) and testing (milliseconds/per sample), as well as the model size (GB) of **AttentionXML with only one PLT** for each dataset. For the ensemble of several trees, AttentionXML can be trained and predicted on a single machine sequentially, or on a distributed settings simultaneously and efficiently (without any network communication).

## 4 Conclusion

We have proposed a new label tree-based deep learning model, AttentionXML, for XMTC, with two distinguished features: the multi-label attention mechanism, which allows to capture the important parts of texts most relevant to each label, and a shallow and wide PLT, which allows to handle millions of labels efficiently and effectively. We examined the predictive performance of AttentionXML, comparing with eight state-of-the-art methods over six benchmark datasets including three extreme-scale datasets. AttentionXML outperformed all other competing methods over all six datasets, particularly datasets with long texts. Furthermore, AttentionXML revealed the performance advantage in predicting long tailed labels.

## Acknowledgments

S. Z. is supported by National Natural Science Foundation of China (No. 61572139 and No. 61872094) and Shanghai Municipal Science and Technology Major Project (No. 2017SHZDZX01). R. Y., Z. Z., Z. W., S. Y. are supported by the 111 Project (NO. B18015), the key project of Shanghai Science & Technology (No. 16JC1420402), Shanghai Municipal Science and Technology Major Project (No. 2018SHZDZX01) and ZJLab. H.M. has been supported in part by JST ACCEL [grant number JPMJAC1503], MEXT Kakenhi [grant numbers 16H02868 and 19H04169], FiDiPro by Tekes (currently Business Finland) and AIPSE by Academy of Finland.

## Footnotes

[1]http://www.ke.tu-darmstadt.de/resources/eurlex/eurlex.html

[2]http://manikvarma.org/downloads/XC/XMLRepository.html

[3] `https://s.yimg.jp/dl/docs/research_lab/annexml-0.0.1.zip`

[4] `https://sites.google.com/site/rohitbabbar/dismec`

[5] `https://github.com/JinseokNam/mlc2seq.git`

[6] `https://github.com/mwydmuch/extremeText`

[7] `https://github.com/xmc-aalto/bonsai`

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
