[Supplementary Material · AttentionXML_NIPS2019_Appendix.pdf]

# Appendix for AttentionXML

**Ronghui You**[1], **Zihan Zhang**[1], **Ziye Wang**[2], **Suyang Dai**[1],
**Hiroshi Mamitsuka**[4,5], **Shanfeng Zhu**[1,3,*]
[1] Shanghai Key Lab of Intelligent Information Processing, School of Computer Science,
[2] Centre for Computational Systems Biology, School of Mathematical Sciences,
[3] Shanghai Institute of Artificial Intelligence Algorithms and ISTBI,
Fudan University, Shanghai, China;
[4] Bioinformatics Center, Institute for Chemical Research, Kyoto University, Japan;
[5] Department of Computer Science, Aalto University, Espoo and Helsinki, Finland
{rhyou18,zhangzh17,zywang17,sydai16}@fudan.edu.cn
mami@kuicr.kyoto-u.ac.jp, zhusf@fudan.edu.cn

## A  Examples of PLT in AttentionXML

Here we show PLTs we used in AttentionXML for three extreme-scale datasets:

1. For Amazon-670K with the number of labels $L = 670,091$, we used a setting of $H = 3$ and $K = M = 2^3 = 8$, the number of nodes in each level of the PLT we used from top to down is 1; 2,048($2^{11}$); 16,384($2^{14}$); 131,072($2^{17}$) and 670,091, respectively.

2. For Wiki-500K with the number of labels $L = 501,008$, we used a setting of $H = 1$ and $K = M = 2^6 = 64$, the number of nodes in each level of the PLT we used from top to down is 1; 8,192($2^{13}$) and 501,008, respectively.

3. For Amazon-3M with the number of labels $L = 2,812,281$, we used a setting of $H = 3$ and $K = M = 2^3 = 8$, the number of nodes in each level of the PLT we used from top to down is 1; 8,192($2^{13}$); 65,536($2^{16}$); 524,288($2^{19}$) and 2,812,281, respectively.

## B  Algorithms

**Algorithm 1** presents the pseudocode for compressing a deep PLT to a shallow one. The deep PLT can be generated by a hierarchical KMeans (K=2) following Parabel [7]. **Algorithm 2** presents the pseudocode for getting labels of tree nodes for each sample. **Algorithm 3** and **Algorithm 4** presents the pseudocode of training and prediction of AttentionXML, respectively.

## C  Related Work

Existing work for XMTC can be categorized into the following four types: 1) 1-vs-All, 2) Embedding-based, 3) Tree-based, and 4) Deep learning-based methods.

### C.1  1-vs-All Methods

1-vs-All methods, such as 1-vs-All SVM, train a classifier (e.g. SVM) for each label independently. A clear weak point is that its computational complexity is very high, and the model size can be huge, due to the extremely large number of labels and instances.

For reducing the complexity, PD-Sparse [12] and PPDSparse [11] are recently proposed by using the idea of sparse learning. PD-Sparse trains a classifier for each label by a margin-maximizing loss function with the $L_1$ penalty to obtain an extremely sparse solution both in primal and dual,

---

**Algorithm 1** Compressing into a shallow and wide PLT

---

**Input:** (a) Labels of training data $\{y_i\}_{i=1}^{N_{train}}$; (b) PLT $T_0$; (c) $K = 2^c$, $H$
**Output:** A compressed shallow and wide PLT $T$

1:   $S_0 \leftarrow \{\text{parent nodes of leaves}\}$
2: **for** $h \leftarrow 1$ **to** $H$ **do**
3:      **if** h < H **then**
4:         $S_h \leftarrow \{c\text{-th ancestor node } n \text{ of nodes in } S_{h-1}\}$
5:      **else**
6:         $S_h \leftarrow \{\text{ the root of } T_0\}$
7:      $T_h \leftarrow T_{h-1}$
8:      **for all** nodes $n \in S_h$ **do**
9:         **for all** nodes $n' \in S_{h-1}$ and node $n$ is the ancestor of $n'$ in $T$ **do**
10:            $Pa(n') \leftarrow n$     ▷ Let node $n$ be parent of node $n'$ in $T_h$, $Pa(n)$ means parent of $n$.
11: **return** $T_H$

---

---

**Algorithm 2** Getting labels of tree nodes

---

**Input:** (a) Labels of training data $\{y_i\}_{i=1}^{N_{train}}$; (b) PLT $T$
**Output:** Tree nodes labels $\{z_i\}_{i=1}^{N_{train}}$

1:   **for** $i \leftarrow 1$ **to** $N$ **do**
2:      **for all** node $n$ in $T$ **do**
3:         $z_{i,n} \leftarrow 0$
4:      **for** $j \leftarrow 1$ **to** $L$ **do**
5:         **if** $y_{i,j}$ **then**
6:            $n \leftarrow$ leaf node corresponding to label $j$ in $T$
7:            **while** $n$ isn't the root of $T$ **do**
8:               $z_{i,n} \leftarrow 1$
9:               $n \leftarrow Pa(n)$
10: **return** $\{z_i\}_{i=1}^{N_{train}}$

---

---

**Algorithm 3** Training of AttentionXML

---

**Input:** (a) Training data $\{x_i, z_i\}_{i=1}^{N_{train}}$; (b) PLT $T$; (c) Candidates number $C$;
**Output:** Trained Model AttentionXML$_1$, AttentionXML$_2$, ..., AttentionXML$_{H+1}$

1:   $H \leftarrow$ the height of $T$
2: **for** $i \leftarrow 1$ to $N$ **do**
3:     $c_i^1 \leftarrow \{$ children node $n$ of the root of $T\}$
4: **for** $d \leftarrow 1$ **to** $H + 1$ **do**
5:      **if** $d > 1$ **then**
6:         **for** $i \leftarrow 1$ **to** $N$ **do**
7:            **for all** node $n \in c_i^{d-1}$ **do**
8:               $\hat{z}_{i,n} \leftarrow$ score predicted by AttentionXML$_{d-1}$ with $x_i$
9:               $\hat{z}_{i,n} \leftarrow \hat{z}_{i,n} \times \hat{z}_{i,Pa(n)}$
10:            $s_i^d \leftarrow$ Top $C$ nodes in $c_i^{d-1}$ by $z_i$ from positive to negative and $\hat{z}_i$ from large to small
11:            $c_i^d \leftarrow \{$ All children nodes of $s_i^d$ in $T\}$
12:         Initialize weights of AttentionXML$_d$ with weights of AttentionXML$_{d-1}$
13:      Train AttentionXML$_d$ with $\{x_i, z_i, c_i^d\}_{i=1}^{N_{train}}$
14: **return** AttentionXML$_1$, AttentionXML$_2$, ..., AttentionXML$_{H+1}$

---

---

**Algorithm 4** Prediction of AttentionXML

---

**Input:** (a) Test sample $x$; (b) PLT $T$; (c) Candidates number $C$;
**Output:** Ranked predicted labels

 1: $H \leftarrow$ the height of $T$
 2: $c^1 \leftarrow \{$ All children nodes the root of $T \}$
 3: **for** $d \leftarrow 1$ **to** $H + 1$ **do**
 4:     **if** $d > 1$ **then**
 5:         $c^d \leftarrow \{$ All children nodes of $s^{d-1}$ in $T \}$
 6:     **for all** node $n \in c^d$ **do**
 7:         $\hat{z}_n \leftarrow$ score predicted by AttentionXML$_d$ with $x$
 8:         $\hat{z}_n \leftarrow \hat{z}_n \times \hat{z}_{Pa(n)}$
 9:     $s^d \leftarrow$ Top $C$ nodes in $c^d$ by $\hat{z}$ from large to small
10: **return** Ranked labels corresponding to $s^{H+1}$

---

without sacrificing the expressive power of the predictor. PPDSparse [11] extends PD-Sparse, by using efficient parallelization of large-scale distributed computing (e.g. 100 cores), achieving a better performance than PD-Sparse.

As another state-of-the-art 1-vs-All method, DiSMEC [1], learns a linear classifier for each label based on distributed computing. DiSMEC uses a double layer of parallelization to sufficiently exploit computing resource (400 cores), implementing a significant speed-up of training and prediction. Pruning spurious weight coefficients (close to zero), DiSMEC makes the model thousands of times smaller, resulting in reducing the required computational resource to a much smaller size than those by other state-of-the-art methods.

## C.2   Embedding-based Methods

The idea of embedding-based methods is, since the label size is huge, to compress the labels and use the compressed labels for training, and finally, compressed labels are decompressed for prediction. More specifically, given $n$ training instances $(\mathbf{x}_i, \mathbf{y}_i)(i = 1, \ldots, n)$, where $\mathbf{x}_i \in \mathbb{R}^d$ is a $d$-dimensional feature vector and $\mathbf{y}_i \in \{0, 1\}^L$ is an $L$-dimensional label vector. Embedding-based methods compress $\mathbf{y}_i$ into a lower $\hat{L}$-dimensional embedding vector $\mathbf{z}_i$ by $\mathbf{z}_i = f_C(\mathbf{y}_i)$, where $f_C$ is called a compression function. Then embedding-based methods train regression model $f_R$ for predicting embedding vector $\mathbf{z}_i$ with input feature vector $\mathbf{x}_i$. For a given instance with feature vector $\mathbf{x}_i$, embedding-based methods predict its embedding vector $\mathbf{z}_i$ by $\mathbf{z}_i = f_R(\mathbf{x}_i)$ and predict label vector $\hat{\mathbf{y}}_i$ by $\hat{\mathbf{y}}_i = f_D(\mathbf{z}_i)$ where $f_D$ is called a decompression function. A disadvantage is that feature space $X$ and label space $Y$ are projected into a low dimensional space $Z$ for efficiency. As such, some information must be lost through this process, sometimes resulting in only limited success.

The main difference among embedding-based methods is the design of compression function $f_C$ and decompression function $f_D$. For example, the most representative method, SLEEC [3], learns embedding vectors $\mathbf{z}_i$ by capturing non-linear label correlations, preserving the pairwise distance between label vectors, $\mathbf{y}_i$ and $\mathbf{y}_j$, i.e. $d(\mathbf{z}_i, \mathbf{z}_j) \approx d(\mathbf{y}_i, \mathbf{y}_j)$ if $i$ is in the $k$ nearest neighbors of $j$. Regressors $\mathbf{V}$ are then trained to predict embedding label $\mathbf{z}_i = \mathbf{V}\mathbf{x}_i$, and a $k$-nearest neighbor classifier (KNN) is used for prediction. KNN has high computational complexity, so SLEEC uses clusters, into which training instances are embedded. That is, given a test instance, only the cluster into which this instance can be fallen is used for prediction.

AnnexML [9] is an extension of SLEEC, solving the three problems of SLEEC: 1) clustering without labels; 2) ignoring the distance value in prediction (since just KNN is used); and 3) slow prediction. AnnexML generates a KNN graph (KNNG) of label vectors in the embedding space, addressing the above problems, and improves both accuracy and efficiency.

## C.3   Tree-based Methods

Tree-based methods use the idea of (classical) decision tree. They generate a tree by recursively partitioning given instances by features at non-terminal nodes, resulting in a simple classifier at each

leaf with only a few active labels. Also following the idea of random forest, most tree-based methods generate an ensemble of trees, selecting (sampling) a feature subset randomly at each node of the trees. A clear disadvantage of the tree-based method is the low performance, because selection at a node of each tree is just an approximation.

The most representative tree-based method, FastXML [8], learns a hyperplane to split instances rather than to select a single feature. In more detail, FastXML optimizes an nDCG (normalized Discounted Cumulative Gain)-based ranking loss function at each node. An extension of FastXML is PfastreXML [4], which keeps the same architecture as FastXML, and PfastreXML uses a propensity scored objective function, instead of optimizing nDCG. Due to this objective function, PfastreXML makes more accurate tail label prediction over FastXML.

Label tree-based methods are already described in the paper.

### C.4 Deep learning-based Methods

Deep learning-based methods can be divided into two types: sequence-to-sequence (Seq2Seq) learning (S2SL) and discriminative learning-based (DL) methods. As the DL methods are already explained in Introduction, we here focus on S2SL methods. Pioneering approaches of S2SL are MLC2Seq [6], SGM [10], and SU4MLC [5], all of which use an attention based Seq2Seq architecture [2]. This architecture has the input with the representations of source text by an RNN encoder and predicts the labels with another attention based RNN decoder. Also trainable attention parameters in this architecture are the same for all labels (Note that AttentionXML has label-specific attention parameters). The difference from MLC2seq is that SGM considers the label distribution at the last time step in decoder, and SU4MLC uses higher-level semantic unit representations by multi-level dilated convolution. Empirically MLC2Seq is demonstrated to outperform FastXML (tree-based method) in terms of F1 measure. In contrast, SGM and SU4MLC have shown no comparative performance advantages.

## D  Experiments and Results

### D.1  Evaluation Metrics

We chose $P@k$ (Precision at $k$) and $N@k$ (normalized Discounted Cumulative Gain at $k$) as our evaluation metrics for performance comparison, since both $P@k$ and $N@k$ are widely used for evaluation methods for multi-labelclassification problems. $P@k$ is defined as follows:

$$P@k = \frac{1}{k} \sum_{l=1}^{k} \mathbf{y}_{rank(l)} \tag{1}$$

where $\mathbf{y} \in \{0,1\}^L$ is the true binary vector, and $rank(l)$ is the index of the $l$-th highest predicted label. $N@k$ is defined as follows:

$$DCG@k = \sum_{l=1}^{k} \frac{\mathbf{y}_{rank(l)}}{log(l+1)}$$

$$iDCG@k = \sum_{l=1}^{min(k,||\mathbf{y}||_0)} \frac{1}{log(l+1)} \tag{2}$$

$$N@k = \frac{DCG@k}{iDCG@k}$$

$N@k$ is a metric for ranking, meaning that the order of top $k$ prediction is considered in $N@k$ but not in $P@k$. Note that $P@1$ and $N@1$ are the same. We also use $PSP@k$(propensity scored precision at $k$) as our evaluation metric for performance comparison on "tail labels" [4]. $PSP@k$ is defined as follows:

$$PSP@k = \frac{1}{k} \sum_{l=1}^{k} \frac{\mathbf{y}_{\text{rank}(l)}}{\mathbf{p}_{\text{rank}(l)}} \tag{3}$$

where $\mathbf{p}_{\text{rank}}(l)$ is the propensity score [4] of label rank$(l)$.

Figure 1: Attention of a test instance (wiki entry: GMail Driver) to the label "gmail" in Wiki10-31K.

## D.2 Performance Results

Table 1 shows the performance comparisons of AttentionXML and other seven state-of-the-art methods over six benchmark datasets.

## D.3 Impact of height and maximum cluster size

Table 2 shows how different maximum cluster sizes $M(= K)$ effect the performance of AttentionXML. For keeping the number of candidates, we use a corresponding $C$ for different $K$. We can see that the setting of a smaller $M$ achieves a better performance on all datasets, especially on "tail labels". Table 3 shows how different heights $H$ effect the performance of AttentionXML. As shown in Table 3, a smaller $H$ achieves a better performance. However, a setting of a smaller $M$ and a smaller $H$ needs more time cost. So choosing these hyper-parameters is a trade-off between performance and time cost .

# E  Effectiveness of Attention

We show a typical case, to demonstrate the advantage of attention mechanism in AttentionXML. Fig. 1 shows a typical text example from test data of Wiki10-31K. One of its true labels is "gmail", which is ranked at the top by AttentionXML (while at the over 100th without multi-label attention). In Fig. 1, each token is highlighted by its attention score to this label computed by AttentionXML. We can see that AttentionXML gives high scores to "Gmail", "e-mail" and "attachments", which are all relevant to the true label "gmail". This result shows that the attention mechanism of AttentionXML is effective for XMTC.

Table 1: Performance comparisons of AttentionXML and other competing methods over six benchmark datasets.

| Methods | P@1=N@1 | P@3 | P@5 | N@3 | N@5 | PSP@1 | PSP@3 | PSP@5 |
|---|---|---|---|---|---|---|---|---|
| | | | EUR-Lex | | | | | |
| AnnexML | 79.66 | 64.94 | 53.52 | 68.70 | 62.71 | 33.88 | 40.29 | 43.69 |
| DiSMEC | 83.21 | 70.38 | 58.73 | 73.73 | 67.96 | 38.45 | 46.20 | 50.25 |
| ProXML | 83.41 | 70.97 | 58.94 | 74.23 | 68.16 | *44.92* | 48.37 | 50.75 |
| PfastreXML | 73.13 | 60.16 | 50.54 | 63.51 | 58.71 | 41.68 | 44.01 | 45.73 |
| Parabel | 82.12 | 68.91 | 57.89 | 72.33 | 66.95 | 37.20 | 44.74 | 49.17 |
| Bonsai | 82.30 | 69.55 | 58.35 | 72.97 | 67.48 | 37.33 | 45.40 | 49.92 |
| XML-CNN | 75.32 | 60.14 | 49.21 | 63.95 | 58.11 | 32.41 | 36.95 | 39.45 |
| AttentionXML-1 | *85.49* | *73.08* | *61.10* | *76.37* | *70.49* | 44.75 | *51.29* | *53.86* |
| AttentionXML | **87.12** | **73.99** | **61.92** | **77.44** | **71.53** | **44.97** | **51.91** | **54.86** |
| | | | Wiki10-31K | | | | | |
| AnnexML | 86.46 | 74.28 | 64.20 | 77.14 | 69.44 | 11.86 | 12.75 | 13.57 |
| DiSMEC | 84.13 | 74.72 | 65.94 | 76.96 | 70.33 | 10.60 | 12.37 | 13.61 |
| ProXML | 85.25 | 76.53 | 67.33 | 78.66 | 71.77 | 17.17 | 16.07 | 16.38 |
| PfastreXML | 83.57 | 68.61 | 59.10 | 72.00 | 64.54 | **19.02** | **18.34** | **18.43** |
| Parabel | 84.19 | 72.46 | 63.37 | 75.22 | 68.22 | 11.69 | 12.47 | 13.14 |
| Bonsai | 84.52 | 73.76 | 64.69 | 76.27 | 69.37 | 11.85 | 13.44 | 14.75 |
| XML-CNN | 81.42 | 66.23 | 56.11 | 69.78 | 61.83 | 9.39 | 10.00 | 10.20 |
| AttentionXML-1 | *87.05* | *77.78* | *68.78* | *79.94* | *73.19* | *16.20* | *17.05* | *17.93* |
| AttentionXML | **87.47** | **78.48** | **69.37** | **80.61** | **73.79** | 15.57 | 16.80 | 17.82 |
| | | | AmazonCat-13k | | | | | |
| AnnexML | 93.54 | 78.36 | 63.30 | 87.29 | 85.10 | 51.02 | 65.57 | 70.13 |
| DiSMEC | 93.81 | 79.08 | 64.06 | 87.85 | 85.83 | 51.41 | 61.02 | 65.86 |
| ProXML | 89.28 | 74.53 | 60.07 | 82.83 | 80.75 | 61.92 | 66.93 | 68.36 |
| PfastreXML | 91.75 | 77.97 | 63.68 | 86.48 | 84.96 | **69.52** | **73.22** | 75.48 |
| Parabel | 93.02 | 79.14 | 64.51 | 87.70 | 85.98 | 50.92 | 64.00 | 72.10 |
| Bonsai | 92.98 | 79.13 | 64.46 | 87.68 | 85.92 | 51.30 | 64.60 | 72.48 |
| XML-CNN | 93.26 | 77.06 | 61.40 | 86.20 | 83.43 | 52.42 | 62.83 | 67.10 |
| AttentionXML-1 | *95.65* | *81.93* | *66.90* | *90.71* | *89.01* | 53.52 | *68.73* | *76.26* |
| AttentionXML | **95.92** | **82.41** | **67.31** | **91.17** | **89.48** | *53.76* | 68.72 | **76.38** |
| | | | Amazon-670K | | | | | |
| AnnexML | 42.09 | 36.61 | 32.75 | 38.78 | 36.79 | 21.46 | 24.67 | 27.53 |
| DiSMEC | 44.78 | 39.72 | 36.17 | 42.14 | 40.58 | 26.26 | 30.14 | 33.89 |
| ProXML | 43.37 | 38.58 | 35.14 | 40.93 | 39.45 | **30.31** | 32.31 | 34.43 |
| PfastreXML | 39.46 | 35.81 | 33.05 | 37.78 | 36.69 | 29.30 | 30.80 | 32.43 |
| Parabel | 44.91 | 39.77 | 35.98 | 42.11 | 40.33 | 26.36 | 29.95 | 33.17 |
| Bonsai | 45.58 | 40.39 | 36.60 | 42.79 | 41.05 | 27.08 | 30.79 | 34.11 |
| XML-CNN | 33.41 | 30.00 | 27.42 | 31.78 | 30.67 | 17.43 | 21.66 | 24.42 |
| AttentionXML-1 | *45.66* | *40.67* | *36.94* | *43.04* | *41.35* | 29.30 | *32.36* | *35.12* |
| AttentionXML | **47.58** | **42.61** | **38.92** | **45.07** | **43.50** | *30.29* | **33.85** | **37.13** |
| | | | Wiki-500K | | | | | |
| AnnexML | 64.22 | 43.12 | 32.76 | 54.30 | 52.23 | 23.98 | 28.31 | 31.35 |
| DiSMEC | 70.21 | 50.57 | 39.68 | 61.77 | 60.01 | 27.42 | 32.95 | 36.95 |
| PfastreXML | 56.25 | 37.32 | 28.16 | 47.14 | 45.05 | **32.02** | 29.75 | 30.19 |
| Parabel | 68.70 | 49.57 | 38.64 | 60.57 | 58.63 | 26.88 | 31.96 | 35.26 |
| Bonsai | 69.26 | 49.80 | 38.83 | 60.99 | 59.16 | 27.46 | 32.25 | 35.48 |
| AttentionXML-1 | *75.07* | *56.49* | *44.41* | *67.81* | *65.77* | 30.05 | 37.31 | 41.74 |
| AttentionXML | **76.95** | **58.42** | **46.14** | **70.04** | **68.23** | *30.85* | **39.23** | **44.34** |
| | | | Amazon-3M | | | | | |
| AnnexML | *49.30* | 45.55 | 43.11 | 46.79 | 45.27 | 11.69 | 14.07 | 15.98 |
| PfastreXML | 43.83 | 41.81 | 40.09 | 42.68 | 41.75 | **21.38** | **23.22** | **24.52** |
| Parabel | 47.42 | 44.66 | 42.55 | 45.73 | 44.54 | 12.80 | 15.50 | 17.55 |
| Bonsai | 48.45 | 45.65 | 43.49 | 46.78 | 45.59 | 13.79 | 16.71 | 18.87 |
| AttentionXML-1 | 49.08 | *46.04* | *43.88* | *47.17* | *45.91* | 15.15 | 17.75 | 19.72 |
| AttentionXML | **50.86** | **48.04** | **45.83** | **49.16** | **47.94** | *15.52* | *18.45* | *20.60* |

Table 2: Performance comparisons of different $M = K$(with corresponding $C$) for AttentionXML.

| Methods | P@1=N@1 | P@3 | P@5 | N@3 | N@5 | PSP@1 | PSP@3 | PSP@5 |
|---|---|---|---|---|---|---|---|---|
| Amazon-670K, $H = 2$ | | | | | | | | |
| $K = 8, C = 160$ | **45.74** | **40.92** | **37.12** | **43.26** | **41.53** | **29.32** | **32.50** | **35.18** |
| $K = 16, C = 80$ | 45.13 | 40.35 | 36.60 | 42.64 | 40.95 | 28.90 | 32.02 | 34.67 |
| $K = 32, C = 40$ | 44.72 | 39.98 | 36.15 | 42.29 | 40.52 | 28.80 | 31.79 | 34.22 |
| $K = 64, C = 20$ | 44.06 | 39.00 | 35.07 | 41.32 | 39.43 | 28.36 | 30.92 | 33.06 |
| $K = 128, C = 10$ | 42.96 | 37.69 | 33.51 | 39.99 | 37.85 | 27.27 | 29.46 | 31.17 |
| Wiki-500K, $H = 1$ | | | | | | | | |
| $K = 64, C = 15$ | **75.07** | **56.49** | **44.41** | **67.81** | **65.77** | **30.47** | **37.27** | **41.69** |
| $K = 128, C = 8$ | 74.88 | 56.16 | 43.93 | 67.46 | 65.21 | 30.16 | 36.92 | 41.05 |
| $K = 256, C = 4$ | 74.26 | 55.06 | 42.30 | 66.29 | 63.39 | 30.22 | 35.87 | 38.95 |
| Amazon-3M, $H = 3$ | | | | | | | | |
| $K = 8, C = 160$ | **49.08** | **46.04** | **43.88** | **47.17** | **45.91** | **15.15** | **17.75** | **19.72** |
| $K = 16, C = 80$ | 48.63 | 45.64 | 43.45 | 46.76 | 45.48 | 15.02 | 17.59 | 19.50 |

Table 3: Performance comparisons of different $H$ for AttentionXML on Amazon-670K.

| Methods | P@1=N@1 | P@3 | P@5 | N@3 | N@5 | PSP@1 | PSP@3 | PSP@5 |
|---|---|---|---|---|---|---|---|---|
| Amazon-670K, $K = 8, C = 160$ | | | | | | | | |
| $H = 2$ | **45.74** | **40.92** | **37.12** | **43.26** | **41.53** | **29.32** | **32.50** | **35.18** |
| $H = 3$ | 45.66 | 40.67 | 36.94 | 43.04 | 41.35 | 29.30 | 32.36 | 35.12 |
| $H = 4$ | 45.29 | 40.47 | 36.73 | 42.83 | 41.13 | 28.88 | 32.08 | 34.79 |

Table 4: Performance comparisons ($P@5$) of AttentionXML with different $H$ on EUR-Lex, Wiki10-31K and AmazonCat-13K. $H = 0$ means without a PLT.

| AttentionXML | H | EUR-Lex | Wiki10-31K | AmazonCat-13K |
|---|---|---|---|---|
| No PLT | 0 | **61.10** | **68.78** | **66.90** |
| Shallow | 2 | 60.88 | 67.27 | 66.28 |
| Deep | 4 | 60.54 | 65.89 | 65.46 |