[Reviews · NeurIPS 2019]

Reviewer 1



========= I have read the author rebuttal and the other reviews. I maintain my rating however as I pointed out in my review I would request the authors to add more clarity to the method section. ========= This paper presents a label tree and deep learning based method for solving extreme classification problems. Proposed method essentially learns a Parabel like label tree (but shallow and wide) and then instead of using linear classifiers it uses attention based neural network to send training points to the respective nodes. In terms of clarity the paper could certainly be improved, particularly the method section. From the current text it is very hard to figure out what exactly the proposed method is doing. Nevertheless the proposed method outperforms a very hard to beat baseline and hence would be an important contribution to the extreme classification community. Another area where I felt paper could be improved is the explanation of why the method works so well? Is it because the linear classifiers used in parabel weren't powerful enough to correctly classify the points or is it that they simply used bag-of-words features and ignored the semantic meaning of the sentence. Finally it is also not clear to me why the proposed method performs well on tail labels.

Reviewer 2



Originality: This is a very interesting algorithmic contribution. The introduced method gets state-of-the-art results under reasonable computation resources. I was reviewing a former version of this paper for some other conference and have to admit that the new version is significantly improved, mainly because the authors have succeeded to decrease the computational costs of the attention-based deep network by using the probabilistic label trees. Quality: The method is sound and the empirical analysis is of high quality. The paper does not have any theoretical contribution, but it is unnecessary for this kind of contribution. Clarity: The paper in general is clearly written. However, the authors could make a better job in description of the methods: - The tree building method seems to be very simple, but I am not sure whether I have understood all the details. A pseudocode would help a lot. - Similarly, it is not clear enough how the underlying idea of probabilistic label trees have been finally implemented (there is neither a pseudocode in the paper nor the code attached to the submission). It seems that the learning follows a kind of beam search, but this is not clearly stated. Let me underline that this is not necessary for probabilistic label trees. It should be enough to use a given training example only in those nodes for which the parent node is positive (this is what the conditioning on z_{Pa(i) = 1} in (1) says). The solution used by the authors looks like a kind of additional negative sampling. A careful discussion should be given here. - It is also not entirely clear from Subsection 2.3 how training and prediction are performed. Are the models for each level trained sequentially from top to the bottom levels or they are trained all at once? How does batch training work in this case? Is the C parameter the same for training and inference using beam search? - Notation used by the authors is not systematically introduced (some symbols are defined in captions of figures). This makes the paper not pleasant to read. On the other hand, the details of conducted experiments (results, parameters, hardware, ablation analysis) are well-described. It is only surprising that the authors did not include the results of extremeText. This is a shallow network based on PLTs that significantly outperforms XML-CNN. Indeed, it gets results inferior to Parabel or Bonsai, but this is an online method methodologically more similar to the algorithm introduced by the authors. Minor comments concerning clarity are given below: - word(where => word (where - regraded => regarded - Do the outputs of AttentionXML \hat{y} correspond to the node variables z_n? - XML-CNN uses binary cross-entropy loss (not the cross-entropy loss function as suggested by the authors) which is theoretically well-justified as it leads to estimation of marginal probabilities (see, for example, "On label dependence and loss minimization in multi-label classification", MLJ 2012, and the extremeText paper). Significance: AttentionXML achieves better results in terms of precision@k than other state-of-the-art algorithms (including 1-vs-all approaches like DiSMEC, which are very hard to beat) on popular XMLC benchmark datasets. It is worth to underline that the method improves the results significantly not only on precision@1, but also on precision@5. While the proposed approach is still quite costly in training and the approach has many additional hyper-parameters that seem to affect results in a significant way, one should not ignore these outstanding results. The presentation of the paper should be improved, but the contribution deserves publication at NeurIPS.

Reviewer 3



====================== Thanks to the authors for the rebuttal, I have updated my score for the paper. It will be helpful to include the additional experiments done as part of the rebuttal to the final version, along with the references. ================ The paper presents a deep learning method based on attention mechanism for Extreme multi-label Text classification (XMTC) problem. The proposed solution is scalable to datasets with upto 3M labels, and is claimed to achieve state of the art results on precision@k and ndcg@k metrics. Originality - The paper has two main parts - (i) using the attention mechanism for XMTC and (ii) shallow tree for scaling up to large datasets. Even though using the attention mechanism is new for this domain, but the idea of shallow trees is somewhat related to a recently proposed method (Bonsai, [10] in the paper). However one of the main concerns is the following : (a) The process of making the tree shallow by node compression is related to similar ideas in hierarchical classification [1,2] below. In this respect, the reference to these works is missing, and should be included. Quality - Though the paper shows that using shallow trees along with the attention mechanism, state-of-the-art performance is achieved. (a) It is not clear how attention mechanism work as standalone i.e. how does it perform on small datasets when there are no scalability issues, and one does not need the shallow trees. It is, therefore, important to evaluate the attention effect separately without the tree structure, and then study the impact of using that in a shallow tree architecture. Does the accuracy improve upon using the shallow tree? How is it work with deep tree? (b) In comparing on propensity metrics, comparison with another state-of-the-art method [3] (ProXML) is missed out. On some of the datasets, the performance of ProXML is relatively better on these metrics. This comparison should therefore be included in the paper. Clarity - Even though the paper is clear in terms of writing, it would be really beneficial if the authors also provided the code. Significance - Since the results of the paper are significantly better than most state-of-the-art methods, it would be of interest to the community. [1] On Flat versus Hierarchical Classification in Large-Scale Taxonomies, NIPS 2013 [2] Learning taxonomy adaptation in large-scale classification, JMLR 2016 [3] Data scarcity, robustness and extreme multi-label classification, Machine Learning Journal, 2019

[Author Response · NeurIPS 2019]

**About code availability:** We have contacted the program chairs if we can provide source codes in the response, and they tell us that the constraint of 1-page PDF without external links applies to everyone. This means that we are unable to provide the source codes at this stage, due to the constraint of the paper reviewing procedure. However we will definitely submit the source code together with the final version of the paper. Please note that this point was also promised during the process of paper submission.

**To reviewer 1**

1. We will revise the paper carefully by incorporating the suggestions to make the paper clearer.

2. The superiority of AttentionXML comes from two key points: 1) multi-label attention, which captures the most important parts of texts for each label, allows to represent a given document differently for each label. In fact the ablation analysis in Section 3.6 shows that using BiLSTM instead of CNN improves the performance, which is however still much worse than Parabel. On the other hand, using multi-label attention instead of pooling makes the performance the best, outperforming Parabel, Bonsai and DiSMEC. Additionally, in Section 4 of Appendix, we show a typical example to demonstrate the advantage of attention mechanism in AttentionXML. 2) A shallow and wide PLT makes AttentionXML to handle extreme scale data efficiently.

3. The good performance of AttentionXML over tail labels can be attributed to the following two factors. 1) a shallow and wide PLT. In contrast to the deep balanced PLT with a large cluster size in Parabel, the PLT constructed in AttentionXML has a smaller tree height and cluster size. In this case, the chance of grouping unrelated (dissimilar) tail labels into one cluster (meta-label) is very small, which makes the model training of tail labels much easier and more accurate. Tables 2 and 3 in Appendix illustrate the impact of different height $H$ and cluster size $M$ on performance. 2) multi-label attention. Previous methods, such as Parabel and DiSMEC, used only one document representation for all labels, including many unrelated tail labels. It is difficult to satisfy so many unrelated (dissimilar) labels by the same text representation. As we discussed above, multi-label attention can handle this point effectively.

**To reviewer 2**

1. We will add a table on the notations used in our paper, revise the method section and add the pseudocodes on tree building, model training and prediction to make the paper clear and easy to follow.

2. Yes, in fact, our solution is like "a kind of additional negative sampling". By using such additional negative sampling, we can get a more precise approximation of log likelihood than only using nodes with positive parents. We will add detailed discussion on this point.

3. Models are trained and predicted sequentially from top to bottom. After training the current model for level $i$, we generate the candidates for level $i + 1$. The value of parameter $C$ we used in training is the same as the one used in inference. We will emphasize these details in the final version.

4. We will include the result of ExtremeText as another baseline. For clarity, the outputs of AttentionXML $\hat{y}$ correspond to the node variable $z_n$. We also use binary cross-entropy loss (Sorry for the typo of missing "binary").

**To reviewer 3**

1. We will add the three references suggested by the reviewer and ProXML as a competing method in experiments with respect to $PSP@k$ (we have contacted the authors of ProXML and are running ProXML with their generous help. Running ProXML is relatively time consuming, but we can put its result in the final version).

2. Following the reviewer's suggestion, we have examined the performance of AttentionXML under three settings (with shallow tree, deep tree and without PLT (No PLT: standalone attention mechanism)) on three relatively small datasets (Table 1). Note that "No PLT" is equivalent to a tree of only the root with $L$ leaves and we used $K = M = 4$ for the other two settings. The experimental results showed that AttentionXML achieved the best performance under "No PLT" (without PLT) on all these three datasets. Also the performance decreased slightly with a deeper tree on all these datasets. This result implies that PLT is an approximation for achieving better scalability, losing the predictive accuracy slightly. Overall we think that this experiment highlights 1) the importance of multi-label attention mechanism to achieve high accuracy, and 2) that of PLT to achieve model scalability.

Table 1: Performance comparisons ($P@5$) of AttentionXML with different $H$. $H = 0$ means without a PLT.

| AttentionXML | H | EUR-Lex | Wiki10-31K | AmazonCat-13K |
|---|---|---|---|---|
| No PLT | 0 | **61.10** | **68.78** | **66.90** |
| Shallow | 2 | 60.88 | 67.27 | 66.28 |
| Deep | 4 | 60.54 | 65.89 | 65.46 |

[Meta-Review · NeurIPS 2019]

The paper improves the SOTA in extreme classification achieving the difficult feat of outperforming one-vs-all techniques. The authors should follow the reviewers suggestions to improve the clarity of the paper, especially the description of the algorithm. They should also add a discussion as to why their technique is able to improve on the SOTA and provide the additional experimental results they included in the rebuttal.